# A note on the $k$-means clustering for missing data

**Yoshikazu Terada**                                            *terada.yoshikazu.es@osaka-u.ac.jp*
*Graduate School of Engineering Science, The University of Osaka*
*Center for Advanced Integrated Intelligence Research, RIKEN*

**Xin Guan**                                                        *guan.xin.c5@tohoku.ac.jp*
*Graduate School of Information Sciences, Tohoku University*

**Reviewed on OpenReview:** *https: // openreview. net/ forum? id= pcqlTvePXS*

## Abstract

The classical $k$-means clustering algorithm requires complete data and cannot be directly applied when observations contain missing entries. An intuitive and computationally efficient extension addresses this issue by minimizing the $k$-means loss over the observed entries only, a strategy considered in several studies. This method is known as $k$-POD clustering. In this paper, we provide a theoretical analysis of this approach and demonstrate that it is generally inconsistent, even under the missing completely at random (MCAR) assumption. Specifically, we show that the expected loss being minimized asymptotically differs from the original $k$-means objective, leading to biased estimates of cluster centers in the large-sample limit. This highlights a fundamental limitation: the method may fail to recover the true underlying cluster structure, even in settings where $k$-means performs well on fully observed data. Nevertheless, when the missing rate per variable is sufficiently low and the dimensionality is high, the method can still produce stable and practically useful results, making it a viable alternative when the complete-case analysis is ineffective.

## 1 Introduction

The $k$-means clustering is one of the most well-known clustering algorithms, which provides a partition minimizing the sum of within-cluster variances. When a given data matrix has missing entries, most clustering algorithms, including $k$-means clustering, cannot be applied, whereas missing data are common in various real data applications (e.g., Wagstaff & Laidler (2005)). Here, we consider the $k$-means clustering for missing data. There are two common approaches for handling missing entries: *complete-case analysis* and *imputation*, both of which can be used as preprocessing steps prior to the clustering (Himmelspach & Conrad, 2010). However, since we delete all data points with missing entries in complete-case analysis, the number of complete cases could be too small in multivariate data. The imputation approach works well when the assumptions on a hidden probabilistic model are correct, while it is often complicated and its computational cost is high (Lee & Harel, 2022). Another approach is to modify the Euclidean distance used in $k$-means clustering. For example, the partial distance that involves only the observed dimensions is a popular choice (Wagstaff, 2004; Lithio & Maitra, 2018; Datta et al., 2018), the main problem is that the modified measurements for distance may not reflect the true structure based on all dimensions and may not even be a distance measure.

As a natural extension of the $k$-means clustering for missing data, the $k$-POD clustering is proposed by Chi et al. (2016), which can be considered as a special case of the matrix completion issue (e.g., see Jain et al. (2013)). For a data matrix $\boldsymbol{X} = (x_{ij})_{n \times p}$, the set of indexes $\Omega \subset \{1, \ldots, n\} \times \{1, \ldots, p\}$ indicates the observed entries. The projection $\mathcal{P}$ onto an index set $\Omega$ is introduced to replace the missing entries with zero. That is, $[\mathcal{P}_\Omega(\boldsymbol{X})]_{ij} = x_{ij}$ if $(i, j) \in \Omega$, 0 otherwise. Further write a binary matrix $\boldsymbol{U} = (u_{il})_{n \times k}$ for the cluster membership, where $u_{il} = 1$ if $i$th observation belongs to $l$th cluster. The $k$ cluster centers are denoted by a matrix $\boldsymbol{M} = (\mu_{lj})_{k \times p}$, the $l$th row of which represents the $l$th cluster center. Then, the loss

function of the $k$-POD clustering is defined as

$$\min_{\boldsymbol{U},\boldsymbol{M}} \|\mathcal{P}_\Omega(\boldsymbol{X} - \boldsymbol{U}\boldsymbol{M})\|_F^2 \ \text{ such that } \ \boldsymbol{U} \in \{0,1\}^{n \times k}, \ \text{ and } \ \sum_{l=1}^k u_{il} = 1 \ (i = 1, \dots, n),$$

where $\|\boldsymbol{A}\|_F = \big(\sum_{i=1}^n \sum_{j=1}^p a_{ij}^2\big)^{1/2}$ denotes the Frobenius norm of a matrix $\boldsymbol{A} = (a_{ij})_{n \times p}$. When $\Omega = \{1, \dots, n\} \times \{1, \dots, p\}$, the above loss is equivalent to that of the $k$-means clustering. Therefore, the $k$-POD clustering ignores the missing entries in the $k$-means clustering. The simple and fast majorization-minimization algorithm can solve the optimization of the above loss. In each iteration of the MM algorithm, missing entries are imputed using the center of the cluster to which the corresponding observation belongs, followed by applying $k$-means clustering to the imputed data. In this sense, $k$-POD clustering can be regarded as a natural extension of the $k$-means clustering for missing data. This loss function is also considered in Lithio & Maitra (2018), and they develop a more efficient algorithm. The $k$-POD clustering stably performs even under a large proportion of missingness. The numerical experiments in Chi et al. (2016) show that the $k$-POD clustering works well under various cases. As mentioned in Chi et al. (2016), we should note that the $k$-POD clustering has the common limitations as the $k$-means clustering. It still seems to work well in those settings where the $k$-means clustering works.

Interestingly, Wang et al. (2019) independently proposes the following $k$-means clustering for missing data:

$$\min_{\boldsymbol{Y},\boldsymbol{U},\boldsymbol{M}} \|\boldsymbol{Y} - \boldsymbol{U}\boldsymbol{M}\|_F^2 \ \text{ such that } \ \boldsymbol{Y} \in \mathbb{R}^{n \times p} : \mathcal{P}_\Omega(\boldsymbol{Y}) = \mathcal{P}_\Omega(\boldsymbol{X}), \ \boldsymbol{U} \in \{0,1\}^{n \times k},$$

$$\text{and } \sum_{l=1}^k u_{il} = 1 \ \text{ for all } \ i = 1, \dots, n.$$

This method is identical to the $k$-POD clustering since the optimal solution of this problem should satisfy $\mathcal{P}_{\Omega^c}(\boldsymbol{Y}) = \mathcal{P}_{\Omega^c}(\boldsymbol{U}\boldsymbol{M})$ where $\Omega^c$ is the complement of $\Omega$.

In this paper, unfortunately, we will show that the $k$-POD clustering provides an essentially different partition of the data space with the $k$-means clustering even under the simplest missing mechanism called the missing completely at random (MCAR). More precisely, the estimated partition obtained by $k$-POD clustering may converge to a limit different from that of $k$-means clustering in the large-sample regime. This implies that $k$-POD clustering can fail to recover the true underlying cluster structure, even in settings where $k$-means clustering works well. By contrast, under the MCAR mechanism, the $k$-means clustering with complete cases (i.e., complete-case analysis) shares the same asymptotic limit as that of $k$-means clustering.

To explain this problem, we demonstrate an illustrative example in Figure 1. Grey data points are generated from a two-dimensional Gaussian mixture distribution ($n = 10^4$), and the $i$th row $(x_{i1}, x_{i2})$ is randomly observed with probabilities $(q_1, q_2) = (1/3, 2/3)$, where $q_j$ $(j = 1, 2)$ is the probability of the $j$th dimension being observed. Here, the number of complete cases is approximately 2200. The green dotted line is the cluster boundary of the $k$-mean clustering with complete cases and is almost the same as the cluster boundary of $k$-means with all grey data points. However, the cluster boundary of the $k$-POD clustering (blue dashed line) is completely different from these boundaries. The essential reason for the distortion result of the $k$-POD clustering lies in the difference between the expected losses of the $k$-means and $k$-POD clusterings. As shown later, the expected loss function of the $k$-POD clustering can be written as the weighted sum of the expected losses of the $k$-means clustering using parts of variables.

## 2 Inconsistency of $k$-POD clustering

In this paper, we adopt the notation conventions recommended by TMLR. Specifically, non-random (deterministic) scalars are denoted by italic letters (e.g., $r_j$), and non-random vectors by bold italic letters (e.g., $\boldsymbol{r}$). Random quantities are distinguished by upright fonts (e.g., $\mathrm{r}_j$, $\mathbf{r}$). Let $\mathbf{x}_1, \dots, \mathbf{x}_n$ be a $p$-dimensional independent sample from a population distribution $P$. Write $\mathbf{X} = (\mathrm{x}_{ij})_{n \times p}$. Here, we simply consider the missing completely at random mechanism. Let $\mathrm{r}_{ij} = 1$ if $\mathrm{x}_{ij}$ is observed and $\mathrm{r}_{ij} = 0$ if $\mathrm{x}_{ij}$ is missing. The response indicator vectors $\mathbf{r}_i = (\mathrm{r}_{i1}, \dots, \mathrm{r}_{ip})^\top$ $(i = 1, \dots, n)$ are independently distributed from a multinomial

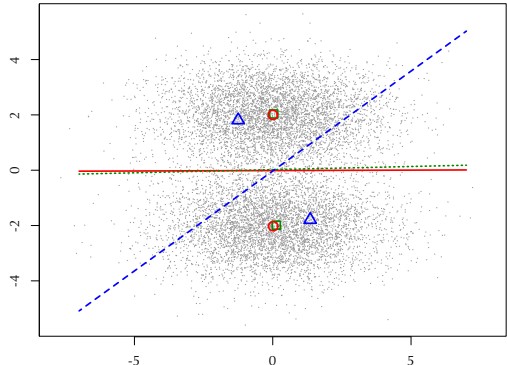

Figure 1: An illustrative example showing that the $k$-POD clustering fails. (solid and circle: $k$-means clustering with all data points, dotted and square: $k$-means clustering with complete cases, triangle and dashed: $k$-POD clustering with missing data).

distribution over the $2^p$ possible missingness patterns and are completely independent with $\mathbf{X}$. We assume that $P(\mathbf{r}_1 = \mathbf{1}_p) > 0$ where $\mathbf{1}_p = (1, \ldots, 1)^\top \in \mathbb{R}^p$.

Let $k$ be the number of clusters. For given cluster centers $\boldsymbol{M} = (\boldsymbol{\mu}_1, \ldots, \boldsymbol{\mu}_k)^\top \in \mathbb{R}^{k \times p}$, the empirical loss of the $k$-means clustering is

$$\widehat{L}_n^{(\text{KM})}(\boldsymbol{M}) = \frac{1}{n} \sum_{i=1}^n \min_{1 \le l \le k} \|\mathbf{x}_i - \boldsymbol{\mu}_l\|^2 = \min_{\boldsymbol{U}} \frac{1}{n} \|\mathbf{X} - \boldsymbol{U}\boldsymbol{M}\|_F^2,$$

where $\boldsymbol{\mu}_l$ is the $l$th row of $\boldsymbol{M}$. Using a similar calculation, the empirical loss of the $k$-POD clustering can be written as

$$\widehat{L}_n^{(\text{KPOD})}(\boldsymbol{M}) = \frac{1}{n} \sum_{i=1}^n \min_{1 \le l \le k} \sum_{j=1}^p \mathrm{r}_{ij}(\mathrm{x}_{ij} - \mu_{lj})^2.$$

Let $\widehat{\mathbf{M}}_{\text{KM}} = \arg\min_{\boldsymbol{M}} \widehat{L}_n^{(\text{KM})}(\boldsymbol{M})$ be the estimator of the $k$-means clustering. Pollard (1981) shows that, as the sample size $n$ goes to infinity, the estimator $\widehat{\mathbf{M}}_{\text{KM}}$ converges to the minimizer of the expected loss of the $k$-means clustering, that is,

$$L^{(\text{KM})}(\boldsymbol{M}) = \mathbb{E}\left[\min_{1 \le l \le k} \|\mathbf{x}_1 - \boldsymbol{\mu}_l\|^2\right].$$

Similarly, we can define the expected loss of the $k$-POD clustering as

$$L^{(\text{KPOD})}(\boldsymbol{M}) = \mathbb{E}\left[\min_{1 \le l \le k} \sum_{j=1}^p \mathrm{r}_{1j}(\mathrm{x}_{1j} - \mu_{lj})^2\right].$$

As the first result, we show that the expected loss of the $k$-POD clustering can be represented as the weighted sum of all possible expected $k$-means losses with parts of variables.

**Proposition 2.1.** *Let $\mathbf{r}_1 = (\mathrm{r}_{11}, \ldots, \mathrm{r}_{1p})^\top$. Under the above assumptions,*

$$L^{(\text{KPOD})}(\boldsymbol{M}) = \sum_{\boldsymbol{r} \in \{0,1\}^p} P(\mathbf{r}_1 = \boldsymbol{r}) L^{(\text{KM})}(\boldsymbol{M} \mid \boldsymbol{r}),$$

*where $\boldsymbol{r} = (r_1, \ldots, r_p)^\top \in \{0, 1\}^p$, and $L^{(\text{KM})}(\boldsymbol{M} \mid \boldsymbol{r})$ is the expected loss of the $k$-means clustering using dimensions with $r_j = 1$:*

$$L^{(\text{KM})}(\boldsymbol{M} \mid \boldsymbol{r}) = \mathbb{E}\left[\min_{1 \le l \le k} \sum_{j=1}^p r_j(\mathrm{x}_{1j} - \mu_{lj})^2\right].$$

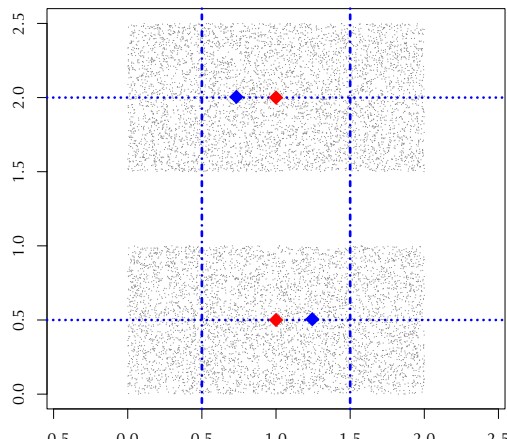

Figure 2: An illustrative example to show why $L^{(\mathrm{KPOD})}$ and $L^{(\mathrm{KM})}$ yield different optimal solutions. (red points: the optimal solution of the $k$-means clustering, blue points: the empirical solution of the $k$-POD clustering with data points).

*Proof.* From the basic property of the conditional expectation, we can immediately obtain

$$L^{(\mathrm{KPOD})}(\boldsymbol{M}) = \mathbb{E}_{\mathbf{r}_1}\left[\mathbb{E}_{\mathbf{x}_1|\mathbf{r}_1}\left[\min_{1\leq l\leq k}\sum_{j=1}^{p}\mathrm{r}_{1j}(\mathrm{x}_{1j}-\mu_{lj})^2 \;\middle|\; \mathbf{r}_1\right]\right] = \sum_{\boldsymbol{r}\in\{0,1\}^p} P(\mathbf{r}_1 = \boldsymbol{r})L^{(\mathrm{KM})}(\boldsymbol{M}\mid\boldsymbol{r}),$$

which completes the proof. $\qquad\square$

Now, we will show the inconsistency of the $k$-POD clustering from the viewpoint of $k$-means clustering. Write $\widehat{\mathbf{M}}_{\mathrm{KPOD}} \in \arg\min_{\boldsymbol{M}} \widehat{L}_n^{(\mathrm{KPOD})}(\boldsymbol{M})$ for an estimator of a cluster center matrix by the $k$-POD clustering. Write $\mathcal{M}^*_{\mathrm{KPOD}}$ for the set of optimal cluster center matrices of the expected loss $L^{(\mathrm{KPOD})}$ (i.e., $\mathcal{M}^*_{\mathrm{KPOD}} = \arg\min_{\boldsymbol{M}} L^{(\mathrm{KPOD})}(\boldsymbol{M})$). We note that $L^{(\mathrm{KPOD})}$ might have multiple optimal solutions even when $L^{(\mathrm{KM})}$ has the unique optimal solution up to relabelling. For example, according to the following theorem, we can assume that the triangular points are close enough to a pair of optimal centers for $L^{(\mathrm{KPOD})}$ in Figure 1. In this case, as this Gaussian mixture is symmetric about the $y$-axis, the pair of points symmetric to the triangular points across the $y$-axis is also near optimal.

The following theorem shows the convergence of the $k$-POD clustering in the large sample limit. The proof is given in the supplementary material.

**Theorem 2.2.** *Assume that $\|\mathbf{x}_1\|$ is bounded almost surely. Then we have, as $n$ goes to infinity,*

$$L^{(\mathrm{KPOD})}\big(\widehat{\mathbf{M}}_{\mathrm{KPOD}}\big) \to \min_{\boldsymbol{M}} L^{(\mathrm{KPOD})}(\boldsymbol{M}) \;\; and \;\; d\big(\widehat{\mathbf{M}}_{\mathrm{KPOD}}, \mathcal{M}^*_{\mathrm{KPOD}}\big) \to 0 \;\; a.s.,$$

*where $d(\boldsymbol{M}, \mathcal{M}^*_{\mathrm{KPOD}}) = \min_{\boldsymbol{M}^*\in\mathcal{M}^*_{\mathrm{KPOD}}} \|\boldsymbol{M} - \boldsymbol{M}^*\|_F$.*

In general, since $L^{(\mathrm{KPOD})}$ and $L^{(\mathrm{KM})}$ have different optimal solutions, $\widehat{\mathbf{M}}_{\mathrm{KPOD}}$ does not converge to an optimal solution of the expected $k$-means loss $L^{(\mathrm{KM})}$. Therefore, the estimated partition of the data space by the $k$-POD clustering is generally different from that by the $k$-means clustering.

**Example 2.1.** *Here, we present a two-dimensional toy example to illustrate why $L^{(\mathrm{KPOD})}$ and $L^{(\mathrm{KM})}$ may yield different optimal solutions. We consider a population distribution that is uniform over the union of two rectangles: $[0.0, 2.0] \times [0.0, 1.0] \cup [0.0, 2.0] \times [1.5, 2.0]$, with the number of clusters set to two. In this case, the optimal solution for $L^{(\mathrm{KM})}$ corresponds to $\boldsymbol{\mu}_1^{(\mathrm{KM})} = (1.0, 0.5)^\top$ and $\boldsymbol{\mu}_2^{(\mathrm{KM})} = (1.0, 2.0)^\top$, shown as red points in Figure 2. We assume that each entry is independently missing with probability 50%. Under this setting, the $k$-POD objective can be expressed as*

$$L^{(\mathrm{KPOD})}(\boldsymbol{M}) = \frac{1}{4}\left\{L^{(\mathrm{KM})}(\boldsymbol{M}\mid(0,1)^\top) + L^{(\mathrm{KM})}(\boldsymbol{M}\mid(1,0)^\top) + L^{(\mathrm{KM})}(\boldsymbol{M}\mid(1,1)^\top)\right\}.$$

*For $L^{(\mathrm{KM})}(\boldsymbol{M} \mid (0,1)^{\top})$, the optimal solutions lie along the dotted line in Figure 2. For $L^{(\mathrm{KM})}(\boldsymbol{M} \mid (1,0)^{\top})$, only the irrelevant variable is observed, and the resulting optimal solutions lie along the dot-dashed line in Figure 2, entirely unrelated to the underlying cluster structure. The loss $L^{(\mathrm{KM})}(\boldsymbol{M} \mid (1,1)^{\top})$ is identical to the k-means loss $L^{(\mathrm{KM})}$. If $L^{(\mathrm{KM})}(\boldsymbol{M} \mid (1,1)^{\top})$ were excluded from the above equation, the optimal centers should be located at the intersection of the dotted and dot-dashed lines. Due to the influence of the irrelevant component $L^{(\mathrm{KM})}(\boldsymbol{M} \mid (1,0)^{\top})$, the k-POD solution is drawn away from the true cluster centers and instead lies at an intermediate position between the k-means solution and the intersection points. In fact, the blue points in Figure 2 show the empirical solution of k-POD clustering with $n = 10^4$, which clearly demonstrates this undesirable phenomenon.*

For high-dimensional data, even if the missing rate of each variable is low, the number of complete cases could be very small. In such cases, the *k*-POD clustering provides much better results than the complete-case analysis. Thus, for high-dimensional data with few missing components, the *k*-POD clustering could be a suitable choice.

## 3 Simulations

In this section, we illustrate some numerical simulations to verify the inconsistency of *k*-POD. We consider the settings in which *k*-means itself can perform well. The Gaussian mixture model $\mathbf{x} \sim \sum_{l=1}^{k} \pi_l N(\boldsymbol{\mu}_l^*, \Sigma_l^*)$ is used to generate the original data, where $N(\boldsymbol{\mu}_l^*, \Sigma_l^*)$ is the *p*-dimensional Gaussian distribution with mean $\boldsymbol{\mu}_l^*$ and covariance $\Sigma_l^*$, and $\pi_l$ is the mixture weight of the *l*th component. The missing completely at random mechanism is considered in this section. We generate the original complete data matrix $\mathbf{X} = (\mathrm{x}_{ij})_{n \times p}$ from the above mixture model and the indicator matrix $\mathbf{R} = (\mathrm{r}_{ij})_{n \times p}$ from the Bernoulli distribution, that is, $\mathrm{r}_{1j}, \ldots, \mathrm{r}_{nj}$ is an independent sample from the Bernoulli distribution with the probability of success $q_j \in (0, 1]$. Then the incomplete data matrix is generated by $\mathbf{X}$ and $\mathbf{R}$, that is, $\mathrm{x}_{ij}$ is observed if $\mathrm{r}_{ij} = 1$ and $\mathrm{x}_{ij}$ is missing if $\mathrm{r}_{ij} = 0$. To measure the bias of the estimator of a cluster center matrix by the *k*-POD clustering $\widehat{\mathbf{M}}_{\mathrm{KPOD}}$, we take the mean square error between $\widehat{\mathbf{M}}_{\mathrm{KPOD}}$ and $\boldsymbol{M}_{\mathrm{KM}}^*$ to be the criterion, which is given by

$$\mathrm{MSE}(\widehat{\mathbf{M}}_{\mathrm{KPOD}}, \boldsymbol{M}_{\mathrm{KM}}^*) = \sum_{l=1}^{k} \min_{l'=1,\ldots,k} \|\hat{\boldsymbol{\mu}}_{\mathrm{KPOD},l} - \boldsymbol{\mu}_{\mathrm{KM},l'}^*\|^2,$$

where $\hat{\boldsymbol{\mu}}_{\mathrm{KPOD},l}$ and $\boldsymbol{\mu}_{\mathrm{KM},l}^*$ are the *l*th rows of $\widehat{\mathbf{M}}_{\mathrm{KPOD}}$ and $\boldsymbol{M}_{\mathrm{KM}}^*$, respectively, and the minimization with respect to $l'$ is to eliminate the influence of index permutation. Since $\boldsymbol{M}_{\mathrm{KM}}^*$ is the minimizer of the expected loss of the *k*-means clustering, it is unknown. We here substitute it by the estimator of the *k*-means clustering with the sample size $n = 10^5$. Since the loss function of the *k*-POD clustering is highly non-convex, as with the original *k*-means clustering, we use multiple random initializations and select the solution with the lowest loss value. More precisely, to initialize the cluster centers in the presence of missing values, we proceed as follows. We first compute the column-wise means of the observed entries and use these to impute the missing values in the data matrix. Each missing entry is replaced with the corresponding column mean. We then randomly select *k* data points from the imputed data to serve as the initial cluster centers. If any of the selected points are duplicated (i.e., some centers are identical due to imputation), we add small random noise to each entry of the initial centers to ensure diversity and numerical stability. To mitigate the effect of local minima, we perform 1000 random initializations and retain the solution with the lowest loss. Here, we should note that Chi et al. (2016) provides the R package `kpodclustr` including the implementation of the *k*-POD clustering with a single specific initialization, which often provides poor local minima with higher loss values.

We first examine the inconsistency of the *k*-POD clustering by analyzing the trend of the MSE as the sample size *n* increases, under the setting described in Figure 1. For comparison, we also include the results of *k*-means clustering applied to the complete cases, as well as those of *k*-means clustering applied to the fully observed (original) data. Figure 3 shows that the MSE of *k*-means with complete cases (dotted line) gradually approaches that of *k*-means with all original data (solid line) as *n* increases. In contrast, the MSE of *k*-POD clustering (dashed line) also converges as *n* increases, but not to zero. The significant gap between the dashed and solid lines, therefore, indicates the inconsistency of the *k*-POD clustering.

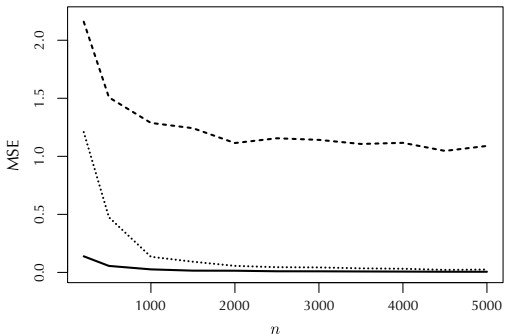

Figure 3: The MSEs of the estimated cluster centers (dashed line: the $k$-POD clustering, solid line: the $k$-means clustering, dotted line: the $k$-means clustering with complete cases).

We further compare the bias of the $k$-POD clustering in estimating cluster centers using several synthetic datasets with $n = 10^4$. The synthetic data are generated from a Gaussian mixture model $\sum_{l=1}^{3} (1/3)\mathcal{N}(\boldsymbol{\mu}_l^*, \boldsymbol{I}_p)$ and then scaled so that each variable has unit sample variance. Missing values are then introduced completely at random (MCAR), with equal probabilities across all variables, based on the fully observed synthetic datasets. We consider the following three settings for the dimensionality and cluster structure of the Gaussian mixture model:

(1) Two-dimensional setting ($p = 2$): The component means are set as $\boldsymbol{\mu}_1^* = (0, 0)^\top$, $\boldsymbol{\mu}_2^* = (3, 0)^\top$, and $\boldsymbol{\mu}_3^* = (1.5, \sqrt{6.75})^\top$.

(2) Five-dimensional setting ($p = 5$): The first two dimensions of the component means are the same as in Setting (1), with the remaining dimensions set to zero.

(3) Fifty-dimensional setting ($p = 50$): Same as Setting (2), except for the number of dimensions.

The missing rate for each variable, denoted by $q$, is set uniformly across variables and takes values in $\{10\%, 30\%, 50\%\}$. All reported values are the means and standard deviations of the MSEs over 100 repetitions. Note that for Setting (3), when the missing rate exceeds 10%, complete-case analysis cannot be performed due to the lack of fully observed samples.

Table 1 summarizes the mean squared errors (MSEs) of different methods across multiple synthetic datasets: *Oracle* (i.e., $k$-means clustering applied to the synthetic data prior to missing value generation), *Complete-case* (i.e., $k$-means clustering applied to complete cases only), and *k-POD* (i.e., $k$-POD clustering applied to incomplete data). It can be observed that, for most datasets, the MSE of the $k$-POD clustering is generally larger than that of the other methods, indicating a substantial bias in the estimated cluster centers when using $k$-POD clustering. In high-dimensional settings, even when the missing rate for each variable is low, the number of complete cases may be too small, rendering complete-case analysis less effective. As shown in the last row of Table 1, $k$-POD clustering performs well, whereas complete-case analysis fails due to the insufficient number of fully observed samples. Therefore, in such scenarios, $k$-POD clustering tends to be more stable and may be more suitable for practical applications.

To further illustrate the behavior of $k$-POD clustering, we consider a well-separated case in which the distances between cluster centers are set to 5 before scaling in Setting (3). Figure 4 presents the estimated cluster centers obtained using $k$-means clustering on the complete data and $k$-POD clustering on the incomplete data. When the missing rate is $q = 10\%$, the centers estimated by $k$-POD clustering are nearly identical to those obtained by $k$-means clustering with the original data. However, when the missing rate increases to $q = 30\%$, the estimated centers by $k$-POD clustering deviate significantly. Since the $k$-POD loss corresponds to a weighted sum of $k$-means losses over all possible subsets of observed variables, the influence of irrelevant features becomes more pronounced as their number increases.

Finally, we present a real data example using the Wine dataset from the UCI Machine Learning Repository (`http://archive.ics.uci.edu/ml`). This dataset contains the results of a chemical analysis of wine samples

Table 1: Mean squared error (MSE) comparison across different methods. Standard deviations are reported in parentheses.

| Setting | $p$ | Miss. rate | Oracle | Complete | $k$-POD |
|---------|-----|------------|--------|----------|---------|
| (1) | 2 | 10% | 0.001 (0.001) | 0.002 (0.001) | 0.011 (0.003) |
| (1) | 2 | 30% | – | 0.003 (0.002) | 0.072 (0.011) |
| (1) | 2 | 50% | – | 0.004 (0.003) | 0.181 (0.023) |
| (2) | 5 | 10% | 0.012 (0.007) | 0.020 (0.014) | 0.161 (0.046) |
| (2) | 5 | 30% | – | 0.070 (0.036) | 1.617 (0.446) |
| (2) | 5 | 50% | – | 0.996 (1.232) | 3.237 (0.661) |
| (3) | 50 | 10% | 0.186 (0.035) | 21.279 (3.012) | 0.782 (1.191) |
| (3) | 50 | 30% | – | NA | 7.584 (0.677) |
| (3) | 50 | 50% | – | NA | 8.662 (0.836) |

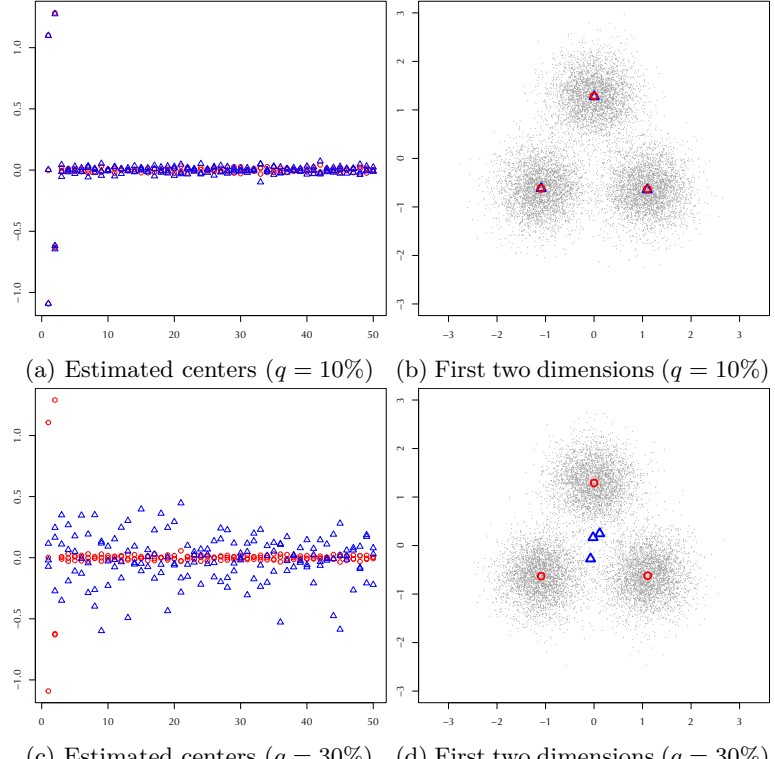

(a) Estimated centers ($q = 10\%$)  (b) First two dimensions ($q = 10\%$)

(c) Estimated centers ($q = 30\%$)  (d) First two dimensions ($q = 30\%$)

Figure 4: Estimated centers (circle: $k$-means with the original data, triangle: $k$-POD with missing data). In (a) and (c), the horizontal axis represents the dimension index, while the vertical axis indicates the center value at each dimension.

produced in the same region of Italy. Although all wines were cultivated under similar environmental conditions, they originate from three different grape cultivars. In our analysis, we treat these three cultivars as the ground-truth cluster structure. The dataset consists of 178 wine samples, each represented by 13 chemical features. In the following, we use data standardized to have zero mean and unit variance for each variable.

We randomly select 50 samples as a validation set and use the remaining 128 samples for training. It is important to note that the training procedure is unsupervised and does not make use of the true cluster labels. To evaluate performance under missing data scenarios, we randomly introduce missingness into

Table 2: Average misclassification rates (standard deviations in parentheses) on the Wine dataset. Setting (a): original features only. Setting (b): with 50 additional irrelevant features.

| Setting | Miss. rate | Training Error | Validation Error |
|---------|------------|----------------|------------------|
| (a)     | 0%         | 0.036 (0.012)  | 0.036 (0.021)    |
|         | 10%        | 0.049 (0.017)  | 0.040 (0.023)    |
|         | 30%        | 0.093 (0.024)  | 0.044 (0.026)    |
| (b)     | 0%         | 0.068 (0.022)  | 0.070 (0.030)    |
|         | 10%        | 0.085 (0.028)  | 0.079 (0.034)    |
|         | 30%        | 0.195 (0.061)  | 0.133 (0.067)    |

the training data at fixed rates (10% and 30%). We then apply the $k$-POD clustering algorithm to the incomplete training data and compute the misclassification rates on both the training and validation sets. This procedure is repeated 100 times.

To assess performance in high-dimensional settings with irrelevant features, we additionally generate independent 50 noise variables from a standard normal distribution and append them to the original dataset. We then repeat the same procedure and compare the resulting accuracies. The average misclassification rates are summarized in Table 2. For reference, we also report the clustering performance of the classical $k$-means algorithm applied to the original dataset, without any artificially introduced missing values (i.e., the 0% missingness cases in Table 2). As shown in Table 2, the results exhibit a similar pattern to those observed in the previous simulation study. These findings suggest that $k$-POD clustering may be unreliable when the missing rate is high, and should, therefore, be applied with caution in such scenarios.

## 4 Conclusions

In this paper, we have examined the theoretical properties of the $k$-means clustering in the presence of missing data, with a particular emphasis on the $k$-POD method, a natural and widely used extension of the classical $k$-means algorithm for missing data. We have shown that the $k$-POD clustering is theoretically inconsistent, even under the most favorable assumption that the data are missing completely at random. More precisely, as the sample size tends to infinity, the $k$-POD estimator does not converge to the optimal solution of the original $k$-means objective but rather to the minimizer of a weighted sum of $k$-means losses computed over subsets of variables defined by the missingness patterns.

This theoretical insight reveals a fundamental limitation of the $k$-POD approach: it may fail to recover the underlying cluster structure, even when the classical $k$-means clustering performs well on fully observed data. Nonetheless, the $k$-POD clustering remains effective, specifically when the missing rate in each variable is sufficiently low and the data are high-dimensional.

These findings highlight the necessity of a rigorous understanding of the theoretical behavior of clustering algorithms in the presence of missing data. We emphasize the importance of exercising caution when applying $k$-means-type methods to incomplete datasets, even when their formulations appear intuitive or computationally appealing.

Moreover, beyond the immediate implications for practitioners using $k$-POD clustering, our findings also highlight fundamental challenges in extending $k$-means-type algorithms to settings with missing data. For highly incomplete data or incomplete data with a complex missing mechanism, we strongly recommend using model-based clustering methods (e.g., Gaussian mixture models) within a likelihood-based framework (Little & Rubin, 2002). The demonstrated inconsistency of $k$-POD may guide the development and analysis of future clustering methods under incomplete data scenarios.

**Acknowledgments**

This research was supported by JSPS, Japan KAKENHI Grant (JP20K19756, JP20H00601, and JP24K14855 to YT), the MEXT Project for Seismology Toward Research Innovation with Data of Earthquakes (STAR-E, JPJ010217 to YT), and China Scholarship Council (NO. 202108050077 to XG). The authors wish to express our thanks to Ms. Jiayu Li for her helpful discussions.

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

## A    Proof of Theorem 1

The proof is similar to the proofs for the consistency of the classical $k$-means. For simplicity of notation, we omit the superscript (KPOD) from $\widehat{L}_n^{(\text{KPOD})}(\boldsymbol{M})$ and $L^{(\text{KPOD})}(\boldsymbol{M})$ through the appendix. Let $C > 0$ be a positive constant such that $\|\mathbf{x}_1\| < C$ almost surely. Let $\mathcal{B}(C) := \{\boldsymbol{x} \in \mathbb{R}^p \mid \|\boldsymbol{x}\| \leq C\}$ be the closed ball. By the $k$-POD algorithm, we can ensure that the estimated centers of the $k$-POD clustering are in $\mathcal{B}(C)$. Define $\mathcal{E}_k(C) = \{\boldsymbol{M} \in \mathbb{R}^{k \times p} \mid \boldsymbol{\mu}_l \in \mathcal{B}(C), \ l = 1 \ldots, k\}$ for $C > 0$.

We write $\widehat{\mathbf{M}}_n \in \arg\min_{\boldsymbol{M} \in \mathcal{E}_k(C)} \widehat{L}_n(\boldsymbol{M})$ for an estimator of a cluster center matrix by the $k$-POD clustering, and write $\mathcal{M}^*$ for the set of optimal cluster center matrices of the expected loss $L$ (i.e.,

$\mathcal{M}^* = \arg\min_{\boldsymbol{M}\in\mathcal{E}_k(C)} L(\boldsymbol{M}))$. Any element of $\mathcal{M}^*$ is denoted by $\boldsymbol{M}^*$. We note that $\boldsymbol{M}^*$ is not necessarily unique.

Since $\mathcal{B}(C)$ is compact, $\mathcal{E}_k(C)$ is also compact under the topology induced by the Hausdorff metric. The following lemma gives the uniform strong law of large numbers and the continuity of the expected loss $L$ on such compact set $\mathcal{E}_k(C)$.

**Lemma A.1.** *Under the assumption in Theorem 1, the followings hold for $C > 0$:*

*(a) The uniform law of large numbers holds:*

$$\lim_{n\to\infty} \sup_{\boldsymbol{M}\in\mathcal{E}_k(C)} \left| \widehat{L}_n(\boldsymbol{M}) - L(\boldsymbol{M}) \right| = 0 \quad a.s., \quad and$$

*(b) The loss function $L(\boldsymbol{M})$ is continuous on $\mathcal{E}_k(C)$.*

Now, we are ready to prove Theorem 1.

*Proof of Theorem 1.* The first result is the immediate consequence from (a) of Lemma A.1. Thus, we will show the second result. By the optimality of $\widehat{\mathbf{M}}_n$, we have

$$\limsup_n \left[ \widehat{L}_n(\widehat{\mathbf{M}}_n) - \inf_{\boldsymbol{M}^*\in\mathcal{M}^*} \widehat{L}_n(\boldsymbol{M}^*) \right] \le 0 \quad \text{a.s.}$$

By the strong law of large numbers,

$$\forall \boldsymbol{M}^* \in \mathcal{M}^*; \; \limsup_n \inf_{\boldsymbol{M}^*\in\mathcal{M}^*} \widehat{L}_n(\boldsymbol{M}^*) \le \limsup_n \widehat{L}_n(\boldsymbol{M}^*) = L(\boldsymbol{M}^*) \quad \text{a.s.}$$

Thus, we obtain

$$0 \ge \limsup_n \widehat{L}_n(\widehat{\mathbf{M}}_n) - \limsup_n \inf_{\boldsymbol{M}^*\in\mathcal{M}^*} \widehat{L}_n(\boldsymbol{M}^*) \ge \limsup_n \widehat{L}_n(\widehat{\mathbf{M}}_n) - L(\boldsymbol{M}^*) \quad \text{a.s.}$$

By the continuity of the expected loss $L$, we have

$$\forall \delta > 0; \; \min_{\boldsymbol{M}\in\mathcal{E}_{k,\delta}(C)} L(\boldsymbol{M}) > \min_{\boldsymbol{M}\in\mathcal{E}_k(C)} L(\boldsymbol{M}),$$

where $\mathcal{E}_{k,\delta}(C) = \{\boldsymbol{M} \in \mathcal{E}_k(C) \mid d(\boldsymbol{M}, \mathcal{M}^*) \ge \delta\}$. Thus, from (a) of Lemma A.1, the above inequality leads that for any $\delta > 0$

$$\liminf_n \inf_{\boldsymbol{M}\in\mathcal{E}_{k,\delta}(C)} \widehat{L}_n(\boldsymbol{M}) \ge \inf_{\boldsymbol{M}\in\mathcal{E}_{k,\delta}(C)} L(\boldsymbol{M}) > L(\boldsymbol{M}^*) \ge \limsup_n \widehat{L}_n(\widehat{\mathbf{M}}_n) \quad \text{a.s.}$$

This gives that there exists $n_0 \in \mathbb{N}$ almost surely such that

$$\forall n \ge n_0; \; \inf_{\boldsymbol{M}\in\mathcal{E}_{k,\delta}(C)} \widehat{L}_n(\boldsymbol{M}) > \widehat{L}_n(\widehat{\mathbf{M}}_n).$$

If $d(\widehat{\mathbf{M}}_n, \mathcal{M}^*) \ge \delta$ for some $n \ge n_0$, we have $\inf_{\boldsymbol{M}\in\mathcal{E}_{k,\delta}(C)} \widehat{L}_n(\boldsymbol{M}) > \widehat{L}_n(\widehat{\mathbf{M}}_n)$, which is impossible. Therefore, we conclude that $\lim_{n\to\infty} d(\widehat{\mathbf{M}}_n, \mathcal{M}^*) = 0$ a.s.

$\square$

# B   Proofs of Lemma 1

Here, we provide the proof of Lemma A.1.

*Proof of Lemma A.1.* For any $\boldsymbol{M} = (\boldsymbol{\mu}_1, \ldots, \boldsymbol{\mu}_k)^\top \in \mathbb{R}^{k \times p}$, define the function $g_{\boldsymbol{M}}(\cdot, \cdot) : \mathbb{R}^p \times \{0, 1\}^p \to \mathbb{R}$ to be $g_{\boldsymbol{M}}(\boldsymbol{x}, \boldsymbol{r}) = \min_{1 \leq l \leq k} \|\boldsymbol{x} \circ \boldsymbol{r} - \boldsymbol{\mu}_l \circ \boldsymbol{r}\|^2$. Let $\mathcal{G} = \{g_{\boldsymbol{M}}(\cdot, \cdot) \mid \boldsymbol{M} \in \mathcal{E}_k(C)\}$.

$$\sup_{g_{\boldsymbol{M}} \in \mathcal{G}} \left| \frac{1}{n} \sum_{i=1}^n g_{\boldsymbol{M}}(\mathbf{x}_i, \mathbf{r}_i) - \mathbb{E}\left[g_{\boldsymbol{M}}(\mathbf{x}_1, \mathbf{r}_1)\right] \right| \to 0 \quad \text{a.s.}$$

From Theorem 19.4 in van der Vaart (2000), it suffices to show that for each $\epsilon > 0$ there exists a finite class $\mathcal{G}_\epsilon$ such that for each $g_{\boldsymbol{M}} \in \mathcal{G}$, there are functions $\mathring{g}_{\boldsymbol{M}}, \bar{g}_{\boldsymbol{M}} \in \mathcal{G}_\epsilon$ with $\mathring{g}_{\boldsymbol{M}} \leq g_{\boldsymbol{M}} \leq \bar{g}_{\boldsymbol{M}}$ and $\mathbb{E}\left[\bar{g}_{\boldsymbol{M}}(\mathbf{x}_1, \mathbf{r}_1) - \mathring{g}_{\boldsymbol{M}}(\mathbf{x}_1, \mathbf{r}_1)\right] < \epsilon$.

For $\delta > 0$, let $D_\delta$ be a finite subset of $\mathcal{B}(C)$ such that

$$\forall \boldsymbol{\mu} \in \mathcal{B}(C); \; \exists \boldsymbol{\nu} \in D_\delta; \; \|\boldsymbol{\mu} - \boldsymbol{\nu}\| < \delta.$$

Define $\mathcal{D}_{k,\delta} = \{\boldsymbol{M} \in \mathcal{E}_k(C) \mid \boldsymbol{\mu}_l \in D_\delta, \; l = 1, \ldots, k\}$. For each $\delta > 0$, we give the finite class $\mathcal{G}_\epsilon$ of the form:

$$\mathcal{G}_\delta = \left\{ \min_{1 \leq l \leq k} \left(\|\boldsymbol{x} \circ \boldsymbol{r} - \boldsymbol{\nu}_l \circ \boldsymbol{r}\| \pm \delta\right)^2 \;\middle|\; \boldsymbol{V} \in \mathcal{D}_{k,\delta}\right\}.$$

For a fixed $\boldsymbol{M} \in \mathcal{E}_k(C)$, take $\boldsymbol{V} = (\boldsymbol{\nu}_1, \ldots, \boldsymbol{\nu}_k)^\top \in \mathbb{R}^{k \times p}$ such that $\boldsymbol{\nu}_l \in D_\delta$ and $\|\boldsymbol{\mu}_l - \boldsymbol{\nu}_l\| < \delta$ for any $l = 1, \ldots, k$. Then for $g_{\boldsymbol{M}} \in \mathcal{G}$, we give the corresponding upper and lower bounds in $\mathcal{G}_\delta$ to be

$$\mathring{g}_{\boldsymbol{M}}(\boldsymbol{x}, \boldsymbol{r}) = \min_{1 \leq l \leq k} \left(\|\boldsymbol{x} \circ \boldsymbol{r} - \boldsymbol{\nu}_l \circ \boldsymbol{r}\| - \delta\right)^2 \; \text{ and } \; \bar{g}_{\boldsymbol{M}}(\boldsymbol{x}, \boldsymbol{r}) = \min_{1 \leq l \leq k} \left(\|\boldsymbol{x} \circ \boldsymbol{r} - \boldsymbol{\nu}_l \circ \boldsymbol{r}\| + \delta\right)^2.$$

We first show that $\mathring{g}_{\boldsymbol{M}} \leq g_{\boldsymbol{M}} \leq \bar{g}_{\boldsymbol{M}}$. Since $\mathring{g}_{\boldsymbol{M}}$ and $\bar{g}_{\boldsymbol{M}}$ are determined by $\boldsymbol{V}$ that satisfies $\|\boldsymbol{\mu}_l - \boldsymbol{\nu}_l\| < \delta$, we have for any $l = 1, \ldots, k$ and $(\boldsymbol{x}, \boldsymbol{r}) \in \mathbb{R}^p \times \{0, 1\}^p$,

$$\|\boldsymbol{x} \circ \boldsymbol{r} - \boldsymbol{\nu}_l \circ \boldsymbol{r}\| - \delta \leq \|\boldsymbol{x} \circ \boldsymbol{r} - \boldsymbol{\mu}_l \circ \boldsymbol{r}\| \leq \|\boldsymbol{x} \circ \boldsymbol{r} - \boldsymbol{\nu}_l \circ \boldsymbol{r}\| + \delta.$$

It follows that $\mathring{g}_{\boldsymbol{M}} \leq g_{\boldsymbol{M}} \leq \bar{g}_{\boldsymbol{M}}$. A simple computation gives

$$\mathbb{E}\left[\bar{g}_{\boldsymbol{M}}(\mathbf{x}_1, \mathbf{r}_1) - \mathring{g}_{\boldsymbol{M}}(\mathbf{x}_1, \mathbf{r}_1)\right]$$

$$= \sum_{\boldsymbol{r} \in \{0,1\}^p} P(\mathbf{r}_1 = \boldsymbol{r}) \cdot \int \left\{\bar{g}_{\boldsymbol{M}}(\boldsymbol{x}, \boldsymbol{r}) - \mathring{g}_{\boldsymbol{M}}(\boldsymbol{x}, \boldsymbol{r})\right\} dP(\boldsymbol{x})$$

$$\leq \sum_{\boldsymbol{r} \in \{0,1\}^p} P(\mathbf{r}_1 = \boldsymbol{r}) \cdot \int \sum_{l=1}^k \left\{\left(\|\boldsymbol{x} \circ \boldsymbol{r} - \boldsymbol{\nu}_l \circ \boldsymbol{r}\| + \delta\right)^2 - \left(\|\boldsymbol{x} \circ \boldsymbol{r} - \boldsymbol{\nu}_l \circ \boldsymbol{r}\| - \delta\right)^2\right\} dP(\boldsymbol{x})$$

$$= 4\delta \sum_{\boldsymbol{r} \in \{0,1\}^p} P(\mathbf{r}_1 = \boldsymbol{r}) \cdot \sum_{l=1}^k \int \|\boldsymbol{x} \circ \boldsymbol{r} - \boldsymbol{\nu}_l \circ \boldsymbol{r}\| dP(\boldsymbol{x}) \leq 4\delta k \left(\int \|\boldsymbol{x}\| dP(\boldsymbol{x}) + C\right).$$

This yields $\mathbb{E}\left[\bar{g}_{\boldsymbol{M}}(\mathbf{x}_1, \mathbf{r}_1) - \mathring{g}_{\boldsymbol{M}}(\mathbf{x}_1, \mathbf{r}_1)\right] < \epsilon$.

Next, we prove the continuity of $L(\boldsymbol{M})$ on $\mathcal{E}_k(C)$. If $\boldsymbol{M}, \boldsymbol{V} \in \mathcal{E}_k(C)$ are chosen to safisfy $\max_{l'} \min_l \|\boldsymbol{\mu}_{l'} - \boldsymbol{\nu}_l\| < \delta$,

$$\forall l \in \{1, \ldots, k\}; \; \exists l'(l) \in \{1, \ldots, k\}; \; \|\boldsymbol{\mu}_{l'(l)} - \boldsymbol{\nu}_l\| < \delta.$$

Moreover, we have

$$
\begin{aligned}
&L(\boldsymbol{M}) - L(\boldsymbol{V}) \\
&= \sum_{\boldsymbol{r} \in \{0,1\}^p} P(\mathbf{r}_1 = \boldsymbol{r}) \cdot \int \left( \min_{1 \le l' \le k} \| \boldsymbol{x} \circ \boldsymbol{r} - \boldsymbol{\mu}_{l'} \circ \boldsymbol{r} \|^2 - \min_{1 \le l \le k} \| \boldsymbol{x} \circ \boldsymbol{r} - \boldsymbol{\nu}_l \circ \boldsymbol{r} \|^2 \right) dP(\boldsymbol{x}) \\
&\le \sum_{\boldsymbol{r} \in \{0,1\}^p} P(\mathbf{r}_1 = \boldsymbol{r}) \cdot \int \max_{1 \le l \le k} \left( \| \boldsymbol{x} \circ \boldsymbol{r} - \boldsymbol{\mu}_{l'(l)} \circ \boldsymbol{r} \|^2 - \| \boldsymbol{x} \circ \boldsymbol{r} - \boldsymbol{\nu}_l \circ \boldsymbol{r} \|^2 \right) \, dP(\boldsymbol{x}) \\
&\le \sum_{\boldsymbol{r} \in \{0,1\}^p} P(\mathbf{r}_1 = \boldsymbol{r}) \cdot \int \max_{1 \le l \le k} \left\{ (\| \boldsymbol{x} \circ \boldsymbol{r} - \boldsymbol{\nu}_l \circ \boldsymbol{r} \| + \delta)^2 - \| \boldsymbol{x} \circ \boldsymbol{r} - \boldsymbol{\nu}_l \circ \boldsymbol{r} \|^2 \right\} \, dP(\boldsymbol{x}) \\
&\le 2\delta \sum_{\boldsymbol{r} \in \{0,1\}^p} P(\mathbf{r}_1 = \boldsymbol{r}) \cdot \int \max_{1 \le l \le k} \| \boldsymbol{x} \circ \boldsymbol{r} - \boldsymbol{\nu}_l \circ \boldsymbol{r} \| \, dP(\boldsymbol{x}) + \delta^2 \le 4C\delta + \delta^2
\end{aligned}
$$

Similarly, we obtain $L(\boldsymbol{V}) - L(\boldsymbol{M}) < 4C\delta + \delta^2$, which completes the proof. $\qquad\square$

