# OpenReview forum: "A note on the $k$-means clustering for missing data"
_TMLR — Accepted by TMLR_

### Review · Reviewer_UsuP · 2025-05-08

**Summary Of Contributions:**

The authors showed that the solution a widely adapted k-means extension to handling missing data concentrates on its population minimal, which in general might not be the same as the original k-means clustering with full data access, even under missing completely at random setup.

**Audience:**

Yes

**Broader Impact Concerns:**

Clustering is an extremely widely encountered task and missing data is more than common. Knowing that a widely used method might not be consistent with full data access method under missing complete at random is an important information for scientists dealing with these data.

**Claims And Evidence:**

Yes

**Requested Changes:**

Major comment
- The authors mentioned after Theorem 2.2 that the optimal solution of $L^{KPOD}$ and $L^{KM}$ are in general different. This needs elaboration (e.g., some intuition on why it is not the same, a citation saying they are under some general condition not the same) or put as an assumption when stating inconsistency e.g., in conclusion. As I understood Thm. 2.2 showed that the k-POD estimator concentrates, so bias is the only way to get inconsistency. I agree with the authors that there is no particular reason that they should be the same but if it is not shown anywhere else the authors should at least provide some evidence this is indeed the case. And I think finite sample simulation is not enough for this purpose since it does not exclude the possibility that the bias vanishes in a very slow rate.

Minor comment
- Please consider keeping track of conditioning in the notations used in and around Prop. 2.1 (e.g. $\mathbb{E}[\dots|\mathbf{r}]$ if you are conditioning). Is there a typo in the second equation in Prop 2.1?

**Strengths And Weaknesses:**

Strength
- It is an interesting observation that k-POD might not be consistent and the theory seems sound.

Weaknesses
- In my opinion there is one missing piece about why the bias in general does not vanish. It is possible it is in literature but I missed it.

---

### Review · Reviewer_599s · 2025-05-09

**Summary Of Contributions:**

The paper shows that the intuitively pleasing k-POD clustering model for k-means like clustering in the context of missingness is inconsistent for the k-means problem even in the MCAR missingness regime. Some experiments are given to illustrate the potential practical implications of this biasedness.

**Audience:**

Yes

**Broader Impact Concerns:**

None apparent

**Claims And Evidence:**

Yes

**Requested Changes:**

As mentioned, I think the paper does a pretty good job as-is in showing what it sets out to. My only request is that the authors attempt to better eliminate the potential for experimental results being too heavily influenced by algorithmic issues and not theoretical ones. A potential for this would be to initialise kPOD with the "oracle" solution for the kmeans. Accepting this is obviously not a possibility in practice, the experiments are trying to illustrate the implications of a theoretical truth, and so I don't think this is a big issue. If kPOD takes a good kmeans solution at initialisation, and turns it into a poor one, this is also strong evidence to show the potential issues with kPOD. I would also suggest the removal of the "real data" example, although I do not feel strongly about this point and so do not "request" it, per se.

**Strengths And Weaknesses:**

Strengths:

- The paper is clear, concise and well presented. The results are not at all surprising, but are nonetheless instructive and worthwhile additions to the literature.

- The experimental set-up is (for the most part) appropriate to demonstrate the main points made by the paper.

Weaknesses:

- The "real data" example doesn't seem to add much, if anything, to the paper. Since there is no methodological contribution being made the simulation experiments do a far better job showing what the authors are presenting.

- Whenever dealing with kmeans clustering problems it is hard to tell if any of the experimental results are just arising because the problems are NP hard. Although I appreciate the authors have explained their approach, they also mention that the implementation of kpod in the R package often leads to poor local minima. It seems more could potentially be done to mitigate the potential that the results arise at least partly due to poor solutions obtained and not poor globally optimal solutions (see below).

---

### Review · Reviewer_5Abs · 2025-05-10

**Summary Of Contributions:**

The authors consider K means clustering with missing values. The previously propsed k-POD calculates distances ignoring missing elements. The main result of the paper is that k-POD does not recover the "true" k-means centroids in the large sample limit. This result is actually relatively straightforward to derive and is backed up by various nice simulated and "real" data results.

**Audience:**

Yes

**Claims And Evidence:**

Yes

**Requested Changes:**

Numerical comparison to imputation in each iteration of k means.

**Strengths And Weaknesses:**

I enjoyed the paper and think it's a nice insight to an important and widely used algorithm. The one thing that is missing imo is a comparison to an EM type strategy for missing data, which seems like the most natural solution to me. EM for the Gaussian mixture model can easily be extended to impute missing values, and the same could be done for k-means by adding a step in each iteration that sets missing values to the centroid value for the cluster the data point is in. It is not obvious to me whether this could be analyzed theoretically (I suspect not), but it would be a valuable addition to the numerical experiments.

Minor comment: don't use "On the other hand" with "on the one hand" first.

---

### Decision · Action_Editor_rZvR · 2025-06-19

**Recommendation:** Accept as is

**Additional Comments:**

N/A.

**Audience:**

Yes

**Audience Explanation:**

This submission focuses on clustering, a fundamental technique widely used in machine learning and various fields of data analysis. The insights presented in this work will be valuable to a broad audience. As one reviewer noted: "Clustering is an extremely widely encountered task and missing data is more than common. Knowing that a widely used method might not be consistent with full data access method under missing complete at random is an important information for scientists dealing with these data."

**Claims And Evidence:**

Yes

**Claims Explanation:**

This submission demonstrates that a commonly used clustering method for handling missing data is not consistent with its counterpart that assumes full data access. The authors provide both theoretical analysis and experimental results to support this claim. Reviewers describe the work as offering “a nice insight into an important and widely used algorithm,” “an instructive and worthwhile addition to the literature,” and “important information for scientists dealing with such data.” One reviewer engaged in a productive discussion with the authors to clarify theoretical aspects, while the others provided comments on the experimental evaluation, which the authors addressed appropriately. Ultimately, all three reviewers recommended acceptance.